# Within-Country Inequality in COVID-19 Vaccination Coverage: A Scoping Review of Academic Literature

**DOI:** 10.3390/vaccines11030517

**Published:** 2023-02-23

**Authors:** Nicole Bergen, Nicole E. Johns, Diana Chang Blanc, Ahmad Reza Hosseinpoor

**Affiliations:** 1Department of Data and Analytics, World Health Organization, 20 Avenue Appia, 1211 Geneva, Switzerland; 2Department of Immunization, Vaccines and Biologicals, World Health Organization, 20 Avenue Appia, 1211 Geneva, Switzerland

**Keywords:** COVID-19, dimension of inequality, disparity, health equity, immunisation, inequality, research, scoping review, vaccination

## Abstract

Since December 2020, COVID-19 vaccines have become increasingly available to populations around the globe. A growing body of research has characterised inequalities in COVID-19 vaccination coverage. This scoping review aims to locate, select and assess research articles that report on within-country inequalities in COVID-19 vaccination coverage, and to provide a preliminary overview of inequality trends for selected dimensions of inequality. We applied a systematic search strategy across electronic databases with no language or date restrictions. Our inclusion criteria specified research articles or reports that analysed inequality in COVID-19 vaccination coverage according to one or more socioeconomic, demographic or geographic dimension of inequality. We developed a data extraction template to compile findings. The scoping review was carried out using the PRISMA-ScR checklist. A total of 167 articles met our inclusion criteria, of which half (*n* = 83) were conducted in the United States. Articles focused on vaccine initiation, full vaccination and/or receipt of booster. Diverse dimensions of inequality were explored, most frequently relating to age (*n* = 127 articles), race/ethnicity (*n* = 117 articles) and sex/gender (*n* = 103 articles). Preliminary assessments of inequality trends showed higher coverage among older population groups, with mixed findings for sex/gender. Global research efforts should be expanded across settings to understand patterns of inequality and strengthen equity in vaccine policies, planning and implementation.

## 1. Introduction

The COVID-19 pandemic has brought substantial attention to matters of health inequality, which is defined as a difference in a measurable aspect of health across socially relevant population subgroups [1]. Health inequalities have been evident in COVID-19 exposure risks, outcomes, responses and impacts and, since the mass rollout of COVID-19 vaccination beginning in December 2020, COVID-19 vaccination coverage [2,3]. In this paper, we review the current state of research about inequalities in COVID-19 vaccination coverage.

The development of vaccines against COVID-19 was a major breakthrough in the scientific world and a turning point for controlling the progression of the pandemic [4,5]. Initially, limited global vaccine supplies meant that vaccination implementation plans prioritised certain population groups. Guidance issued by the World Health Organization (WHO) recommended prioritisation of older adults, health workers and immunocompromised persons [6]. As vaccine supplies have become more widely available, however, the inadequate uptake of vaccines by some populations has limited their potential for impact. As of January 2023, nearly 70% of the global population has received at least one dose of a COVID-19 vaccine, although only one-quarter of people in low-income countries have this level of coverage [7]. Inequalities in COVID-19 vaccination coverage are also evident within countries, as certain population subgroups remain systematically disadvantaged; that is, unvaccinated or under-vaccinated. For instance, there were early indications of racial inequity just weeks after vaccine distribution began in the United States, as available data suggested lower vaccination among Black and Hispanic people alongside higher shares of cases and deaths [8]. More recently, data from 14 million adults across 90 countries suggested pervasive education-related inequalities in self-reported receipt of a COVID-19 vaccine in nearly every country, with higher vaccination among the more educated [9].

Distinct factors contribute to COVID-19 vaccination inequalities between countries versus inequalities within countries [10,11]. In this review, we focus on a growing body of research dedicated to exploring within-country inequalities in COVID-19 vaccination coverage. Broadly, the body of research addresses how coverage varies according to dimensions of inequality (i.e., criteria that define population subgroups, such as age, economic status, education level, place of residence, sex or subnational region) that are relevant within a specified population and context. Research efforts to characterise these inequalities offer important insights into situations that may be inequitable (unfair, unjust and/or avoidable through reasonable means [1]). Namely, assessments of inequalities in vaccination coverage can provide evidence about which population subgroups had access to and received the vaccine, and which did not. This evidence can inform how national policies and programmes may be targeted to reach disadvantaged groups and, when repeated over time, monitoring inequalities can support enhanced accountability for upholding and advancing health equity [12]. 

The present review considers research pertaining to COVID-19 vaccination coverage—the actual receipt or non-receipt of a vaccine. We pose the question: what is the current status of research on within-country inequalities in COVID-19 vaccination coverage? COVID-19 vaccination coverage is defined based on the receipt or non-receipt of a COVID-19 vaccine and/or booster dose. The primary objective of the paper is to describe how within-country inequalities in COVID-19 vaccination coverage have been researched and characterised in the academic literature. Specifically, we seek to understand the settings, populations, vaccination indicators, dimensions of inequality and reporting practices featured in this body of research. A secondary objective is to provide a preliminary narrative overview of the trends in inequalities reported for dimensions of inequality that are most frequently addressed by this body of literature. Our findings will be useful to identify and justify areas for further study on this topic, including the design of more detailed systematic literature reviews and meta-analyses. 

## 2. Methods

To address the research question, we conducted a scoping review. Scoping reviews are appropriate “to determine the scope or coverage of a body of literature on a given topic and give clear indication of the volume of literature and studies available as well as an overview (broad or detailed) of its focus” [13]. Further, scoping reviews are useful for assessing an emerging body of evidence to determine specific avenues for further study, for example, through more focused systematic reviews and meta-analyses [13]. Drawing from guidance in the JBI Manual for Evidence Synthesis [14], we developed a protocol for this scoping review, which was refined throughout the process of the review (Appendix A). No major deviations from the protocol were introduced. In preparing this manuscript, we followed the Preferred Reporting Items for Systematic reviews and Meta-Analyses extension for Scoping Reviews (PRISMA-ScR) checklist [15].

### 2.1. Eligibility Criteria

Our focus was on the state of research on inequalities in COVID-19 vaccination, as characterised in the academic literature. Therefore, research articles and reports with primary or secondary data were considered for inclusion, as well as peer-reviewed pre-prints, brief reports and short research communications. Opinion pieces such as comments, letters and editorials, were excluded along with publications of a journalistic and/or less-academic nature (e.g., news stories, biographies and interviews). Relevant publications pertained to human populations. No language restrictions were applied. For inclusion in the review, the full text of the article needed to be available.

Articles were considered for inclusion if the objective pertained to reporting inequalities in COVID-19 vaccination coverage by one or more dimension of inequality. For the purpose of this review, we considered a broad conceptualisation of COVID-19 vaccination (Table 1). Articles were considered for inclusion if the COVID-19 vaccination coverage indicator was defined based on the receipt or non-receipt of any one or more COVID-19 vaccine and/or booster. To be considered for inclusion, dimensions of inequality could encompass any or multiple socioeconomic, demographic or geographic factors; publications that focused on reporting vaccination coverage according to medical factors were excluded. Our scoping review is focused on within-country inequality; therefore, dimensions of inequality could be measured at the individual, household, community or small-area level. Articles that primarily reported between-country inequality were excluded. 

### 2.2. Search Strategy and Screening Process

A systematic search was conducted on 27 October 2022 in PubMed, Scopus and Web of Science. Additional potential sources were obtained through handsearching reference lists. No language, article type or date restrictions were applied to the search (though due to the nature of the research question, eligible articles were published after the rollout of COVID-19 vaccines, which began in late 2020). The electronic search strategy consisted of three domains related to ‘inequality’ AND ‘COVID-19’ AND ‘vaccines’ search terms, incorporating medical subject heading (MeSH) indexed terms, keywords, topic terms and terms used in the title or abstract (Appendix A). In the case of PubMed, additional searches were conducted using MeSH terms related to ‘COVID-19 vaccines’. 

The results from the literature search were imported to Covidence software for removal of duplicates, title and abstract screening, full text review and data extraction. Title and abstract screening was done by one reviewer, followed by full text review conducted in duplicate by two reviewers. Any discrepancies were resolved through discussion and in consultation with a third reviewer, as needed. For studies excluded during the full text review, the first reason for exclusion was recorded, according to the following ordered list: Wrong article type;Does not pertain to humans;Study objective not relevant;Does not meet criteria for COVID-19 vaccination coverage;Does not meet criteria for dimension of inequality;Only reports on between-country inequality;Full text not available;Insufficient information to assess eligibility.

If the full text of the article was not available online, we requested it from the corresponding authors. Likewise, if we did not have sufficient information to assess the eligibility of the study, we made multiple attempts for clarification from corresponding authors.

### 2.3. Data Extraction and Analysis

Data extraction was performed using the Covidence Extraction 2 tool, based on a custom data extraction template. The template covered general information about the article and where it was published; characteristics of the study setting, population, study objective and design; characteristics of the COVID-19 vaccination indicator; characteristics of the dimensions of inequality; analysis methods; results; and conclusions. After reviewing 10% of studies in tandem to ensure consistency in the interpretation and application of the template, data extraction was performed by one of two reviewers. The reviewers reached consensus on any questions or points of ambiguity that arose with input from a third reviewer.

Using the data extraction outputs, descriptive data analysis was undertaken to assess and tabulate study characteristics. In describing the frequency of dimensions of inequality and inequality trends, we used the PROGRESS-Plus framework as a starting point for grouping dimensions of inequality pertaining to common themes. PROGRESS factors include place of residence, race/ethnicity/culture/language, occupation, gender/sex, religion, education, socioeconomic status and social capital [16]. We also identified the following categories, some of which align with the factors described in the “Plus” component of the above framework: age; disability status; family size or composition; health insurance; housing type or characteristic; marital status; migration status; sexual orientation; subnational region or area; and vulnerability, deprivation or poverty index.

As an extension of our analysis, we assessed the preliminary trends in the findings related to dimensions of inequality that appeared most often in the assessed articles. To this end, we coded and compiled reported findings for age; race, ethnicity, cultural group, language and nationality or country of birth; and sex or gender. For age and sex or gender, where the criteria for measuring the dimension were largely comparable across most studies (as years or male/female, respectively), we coded the main findings according to the directionality of the inequality. For race, ethnicity, cultural group, language and nationality or country of birth, where the criteria for measuring the dimension were heterogeneous, we coded whether inequality related to this dimension was reported as meaningful or not meaningful. Our coding of results as meaningful or not meaningful was based on the conclusions reported in the original studies. Most, but not all, studies defined this as statistically significant in comparisons at *p* < 0.05; however, the nature of statistical comparisons differed by paper and not all papers reported statistical significance. 

## 3. Results

### 3.1. Selection of Sources

Our search identified 7784 items (after removing duplicates) that were considered for inclusion. After screening titles and abstracts, 315 were retained for full-text review. A total of 148 items were excluded at the full-text review stage because the study objective was not relevant, the criteria for COVID-19 vaccination coverage was not met, the article type did not meet the inclusion criteria, the criteria for dimension of inequality was not met, the article only reported between-country inequality or there was insufficient information to assess eligibility. In total, 167 articles were included from which data were extracted (Figure 1). 

The included articles were published in 2021 (43 articles) or 2022 (124 articles), across a total of 82 academic journals (Appendix A). Journals represented by five or more included articles were: *Vaccines* (18 articles); *Morbidity and Mortality Weekly Report* (15 articles); *Vaccine* (11 articles); *International Journal of Environmental Research and Public Health* and *PLoS One* (7 articles each); and *BMJ Open* (5 articles). The articles were of variable lengths and scope/depth of analysis, context and discussion. To provide an indication of the type of articles included, we categorised them as full research papers (around six pages or longer, with greater analytical scope and contextual detail) or short research papers (around five pages or less with more limited analytical and contextual detail). The article type designated by the journal also informed the categorisation. Most included articles were designated as full research papers (119 articles, including 1 review), with the remaining designated as short research papers (48 articles).

### 3.2. Study Characteristics

#### 3.2.1. Setting and Study Populations

Most articles contained data from a single country setting (161 out of 167 articles), representing a total of 38 countries. Eighty-three of the single country studies were conducted in the United States of America, followed by studies in the United Kingdom (13 articles), Israel (7 articles),Canada (6 articles) and Hong Kong (5 articles). Six articles included data from multiple countries (representing a total of 18 countries). In total, the number of unique countries represented across all studies was 47. According to the current World Bank classifications [17], 26 of the 47 countries are high-income countries (55%), 11 are upper-middle-income countries (23%), 6 are lower-middle-income countries (13%), and 3 are low-income countries (6%); the remaining 1, Palestine, is not classified (Table 2).

While many studies drew from a national population, others pertained to one or more subnational administrative areas or specified institutions (such as hospitals, universities and prisons). Noting that some articles included more than one of the populations listed below, study populations included general public/adults (88 articles); health care workers (21 articles); older adults (14 articles); pregnant or postpartum women (12 articles); children/adolescents (10 articles); military personnel/veterans (7 articles); university students and staff (5 articles). A smaller number of articles focused on the following populations: people defined based on migratory status (3 articles); incarcerated people (2 articles); LGBTQ+ people (2 articles); people who inject drugs (1 article); EMTs and paramedics (1 article); teachers/staff at schools (1 article); and nursing home residents and staff (1 article). 

#### 3.2.2. COVID-19 Vaccination Coverage

As per our inclusion criteria, all 167 included studies defined COVID-19 vaccination coverage based on the receipt or non-receipt of any one or more COVID-19 vaccine and/or booster. Many articles focused on vaccine initiation, that is, receipt or non-receipt of at least one dose of vaccine (97 articles). In 29 articles, COVID-19 vaccination coverage was defined as ‘fully vaccinated’ according to the specifications of the study setting, and in 4 articles, the focus was on receipt or non-receipt of a COVID-19 vaccine booster. A total of 33 articles looked at multiple COVID-19 indicators that met our inclusion criteria, and the remaining 4 articles did not clearly state the number of vaccine doses or boosters used to define the receipt or non-receipt of a COVID-19 vaccination.

Information about COVID-19 vaccination coverage was sourced from surveys (84 articles), and administrative or surveillance records, including health records (84 articles), noting that one article used data from both of these types of sources. In some cases, administrative or surveillance data were linked to census data to derive denominator values. COVID-19 vaccination coverage was commonly measured at the level of the individual (138 articles), although some articles presented data aggregated at the small-area level (such as county, municipality, zip-code area, province/state or census area) (27 articles), or by institution (such as nursing home or school) (2 articles). 

#### 3.2.3. Dimensions of Inequality

Articles assessed inequalities in COVID-19 vaccination coverage according to diverse socioeconomic, demographic and/or geographic dimensions of inequality (Table 3). In 157 out of 167 articles, inequality in vaccine coverage was reported for at least two dimensions of inequality. The most common dimension of inequality applied was age (127 articles), followed by dimensions of inequality related to race, ethnicity, cultural group, language and nationality or country of birth (117 articles). Inequalities according to sex or gender were reported in 103 articles and 81 articles reported data disaggregated by occupation- or employment-related factors. Other dimensions of inequality that were featured in 10 or more articles include education (76 articles); subnational region or area (68 articles); economic status (68 articles); place of residence (39 articles); vulnerability, deprivation or poverty index (38 articles); marital status (30 articles); family size or composition (27 articles); health insurance (27 articles); and disability status (10 articles). Religion (8 articles), housing type or characteristic (7 articles), migration status (5 articles), social capital (3 articles) and sexual orientation (3 articles) were included less often. Articles relied on different criteria to define and measure dimensions of inequality, with variation depending on the context of the study.

Information about dimensions of inequality was sourced from surveys (93 articles) and administrative or surveillance data, including health records (79 articles) and censuses (20 articles) (note that 24 articles relied on more than one type of data source). In most articles, dimensions of inequality were measured at the same level as the corresponding COVID-19 vaccination indicator (149 articles). In some articles, various dimensions of inequality measurements included both individual and small-area levels (15 articles). Three articles measured the dimension of inequality at the small-area level and the COVID-19 vaccination indicator at the individual level.

#### 3.2.4. Reporting Practices

In general, articles presented disaggregated data and association or regression measures when reporting inequality findings (108 articles). Thirty-three articles included disaggregated estimates only, while 16 articles reported association or regression measures only. Sixteen articles included difference, ratio, slope index of inequality and/or relative index of inequality summary measures. 

About a third of articles (54 out of 167) reported multiple disaggregation; that is, they presented data about vaccination coverage broken down by two or more dimensions of inequality simultaneously. For example, several articles explored sex- or age-related inequalities across different urban–rural classifications [18,19,20,21].

### 3.3. Study Findings: Preliminary Trends for Selected Dimensions of Inequality

#### 3.3.1. Age

Of the 127 articles that reported COVID-19 vaccination coverage by age, the majority (89 articles, or 70%) found higher vaccination among older groups. The age ranges and categorisation (groupings) of these 89 articles were diverse: while many study populations included those aged 18 years and older, some were limited to other age groupings. For example, a study in England reported higher rates of being unvaccinated in younger individuals of study populations aged 50 years or older [22]. Similarly, vaccination with one or more doses of a COVID-19 vaccine was higher among those aged over 75 years compared to those 65–74 years in Connecticut, United States [23], and higher among those 75 or older compared to those 60–74 years in Sweden [24]. A study by Khatatbeh et al. (2022) in the Eastern Mediterranean Region, looked at COVID-19 vaccination in children aged 12 or younger versus those aged 12–17; it also explored the association between parental age and receipt of a COVID-19 vaccine in their child(ren). In both cases—child age and parental age—older ages were predictive of the child being vaccinated [25]. Studies in other settings, including Indonesia [26] and the United States [27,28] also reported positive associations between age and COVID-19 vaccination coverage within child/adolescent populations.

In contrast, the opposite pattern—higher vaccination among younger groups—was reported in 8 articles (6%). These articles focused on adult populations across different settings, including health care workers in China [29] and Egypt [30]; university students or staff in the United States [31,32]; active military personnel in Israel [33]; pregnant or postpartum women in the United States [34]; and general adult populations in Singapore [35] and the United States [36]. 

Nineteen of the articles that reported on age-related inequality (15%) found no/minimal inequality and 9 articles (7%) demonstrated other patterns (such as higher vaccination coverage in a mid-range age group) or mixed patterns (such as different age-related patterns for different population groups, or for different COVID-19 vaccination indicators). Two articles (2%) did not report age-related findings in the main text of the article. 

#### 3.3.2. Race, Ethnicity, Cultural Group, Language, Nationality or Country of Birth

Overall, 117 articles reported on COVID-19 vaccination coverage by race, ethnicity, cultural group, language, nationality and/or country of birth. Nearly all of these articles (101 articles, or 86%) reported meaningful inequality according to this dimension of inequality, while 18 articles (15%) reported no meaningful inequality (noting that 3 articles included in the above counts reported both meaningful and non-meaningful findings for different variables in this category). One article (1%) did not report the findings for this dimension in the main text of the article.

Most of the studies conducted in the United States included a dimension of inequality related to race, ethnicity, cultural group, language, nationality and/or country of birth (73 out of 83, or 88%), and in 88% of these studies (64 out of 73), authors reported meaningful inequality by at least one of these dimensions. Although standardised racial/ethnic diversity categories used in the United States Census are applied in many studies, it is difficult to assess trends in the findings due to different study designs and comparison groups. We observed, however, that Asian and/or White subgroups were often among the most advantaged with regards to COVID-19 vaccination coverage. For instance, in a study of race/ethnicity inequalities in the United States, subgroups identifying as Asian or White had higher booster uptake than Black and Hispanic populations in all of the states for which there were data (24 states plus Washington, D.C.) [37]. Di Rago et al. (2022), assessing COVID vaccination rates across eight American cities over a three-week period, found increasing gaps in vaccination between White or Asian and Black or Hispanic communities [38]. 

#### 3.3.3. Sex or Gender

A total of 103 articles reported on COVID-19 vaccination coverage by sex or gender, of which 45 articles (44%) found no or minimal difference between subgroups. In 31 articles (30%), COVID-19 vaccination was higher among males; in 25 articles (24%), vaccination was higher among females. Two articles (2%) reported different patterns of sex-related inequality across age groupings [39] or by disability status [40]. 

Two studies, both focusing on LGBT or LGBTQ+ adults in the United States, considered sex and gender as separate variables in their analysis. Low et al. (2022) reported no differences based on sex assigned at birth (categorised as female, male and intersex) or gender identity (categorised as cisgender and transgender/nonbinary/other gender minority) [41]. McNaghten et al. (2022) reported higher vaccination among females than males, but no difference based on gender identity (dichotomously categorised as transgender/nonbinary or not) [42]. 

## 4. Discussion

In this scoping review, we assessed the current state of research pertaining to within-country inequalities in COVID-19 vaccination coverage. Our findings show that this body of research covers a diversity of populations and settings, suggesting a wide interest in assessing and understanding the patterns of vaccination coverage inequality across populations. The geographical representation of study settings within this literature favoured high-income countries in North America and Europe—for example, half of articles were conducted in the United States, with only five articles based on populations in the African continent. High-income countries accounted for more than half of the countries represented in this body of literature, whereas low-income countries accounted for less than 10%. This finding was not surprising, as populations in high-income countries tended to have earlier access to vaccines than lower-income countries, and thus implemented vaccine programmes sooner; moreover, timelines, access and incentives for publishing in academic journals may differ between settings. Nevertheless, more research on inequalities in COVID-19 vaccination coverage is warranted in lower-income countries, particularly as vaccines become more widely available in these settings.

We found that demographic factors, including age, race/ethnicity/cultural group/language/nationality/country of birth and sex/gender, were the most commonly reported dimensions of inequality in vaccine coverage rates in this body of literature. Indeed, early into the COVID-19 pandemic, the scientific community made strong calls to enhance the collection and reporting of data disaggregated by these factors [43,44,45,46]. In the cases of age and sex/gender, the application of similar measurement criteria (years and male/female, respectively) allowed us to comment on the general trends in the directionality of inequality reported in these articles. Our preliminary assessment of this literature suggested that vaccination coverage tended to be higher among (relatively) older population groups, across many age ranges. This is in line with WHO guidance [6], suggesting the implementation of vaccine rollout strategies that initially prioritised older age groups. With regards to findings on sex/gender-related inequality, a substantial proportion of articles that reported on this dimension concluded that there were no meaningful differences. Of those articles that did report a difference, the directionality was variable, with vaccination coverage more often reported to be higher in males than in females. It was not feasible to do even preliminary comparisons of findings for the race/ethnicity/cultural group/language/nationality/country of birth dimension of inequality, as the measurement of this dimension is context specific (i.e., not standardised across settings). There were, however, common approaches applied within particular country settings, which could be explored through more narrowly-focused systematic reviews and/or meta-analyses. Indeed, more rigorous meta-analyses, including quality assessments, are warranted to delve into vaccination coverage inequalities by demographic factors.

Among the other dimensions of inequality highlighted in our scoping review were those related to socioeconomic factors (most frequently occupation/employment, education level, economic status and vulnerability, deprivation or poverty indices) and those related to geographical factors (most frequently subnational region/area and place of residence). Characterising patterns of socioeconomic inequality offers deeper understanding into the motivations and barriers experienced by population groups, while geographic patterns of inequality may have immediate and practical implications for program delivery and resource allocation [47]. Other dimensions of inequality, such as sexual orientation, social capital, and migration status, received less attention in the research, although some of these articles provide initial indications that these may be meaningful avenues for future study. For instance, the three articles that reported inequality in COVID-19 vaccination coverage related to a measure of social capital all reported higher vaccination among groups with greater social capital [20,48,49]. 

The application of multiple disaggregation in one-third of studies permitted exploration of the intersection of different dimensions of inequality. Multiple disaggregation can begin to lend insight into more nuanced patterns of inequality, for example, suggesting how multiple vulnerabilities may put certain groups at heightened risk for lower vaccination coverage [50]. Multiple disaggregation should be incorporated, to the extent possible, in future inequality analyses in this topic [51]. 

We reported variability in how dimensions of inequality were measured, reflecting diverse study populations, settings and research aims. In some cases, standardised criteria were applied within a country (such as race/ethnicity categories in the United States), enhancing comparability across these studies. For most dimensions of inequality, however, the lack of standardised criteria for measuring dimensions of inequality limits the extent to which direct comparisons of inequality can be made across studies and across settings. 

Our scoping review extends on a previous review by Bayati et al. (2022), which had a broader aim of assessing both between-country and within-country factors associated with COVID-19 vaccine distribution [11]. The portion of the review focused on within-country factors included 19 studies, and concluded that “age, race, ethnic, household income, residency in the deprived areas, employment, poverty, location (urban/rural) and gender were most often mentioned in the literature”. Our scoping review, encompassing 167 studies, highlights additional dimensions of inequality that have been explored in the literature, including education level, indices of vulnerability, deprivation or poverty, marital status and family characteristics. Additionally, it provides a more detailed overview of the study settings, data sources, reporting practices and preliminary findings.

The findings of this scoping review are broadly in line with previous reviews exploring inequalities in COVID-19 vaccination intentions and attitudes [52,53]. For instance, the directionality of the age- and sex/gender-related inequality that we reported corresponds with those reported in a meta-analysis on inequalities in COVID-19 vaccination intention, which included 28 nationally representative populations across 13 countries. It reported female sex, younger age and belonging to an ethnic minority group to be consistently associated with lower intention to vaccinate, highlighting “an urgent need to address social inequalities in vaccine hesitancy and promote widespread uptake of vaccines as they become available” [52]. Similarly, a meta-analysis including 63 surveys and more than 30 countries concluded that age, gender and education level were among the factors most often associated with willingness or hesitancy to be vaccinated [53]. We note, however, an important distinction between the body of research pertaining to vaccination coverage from research on vaccination intentions and attitudes. Attitudes towards vaccines have been found to shift over time [9], and do not directly translate into behaviours. A study of vaccination uptake during the 2009 influenza (H1N1) pandemic, for example, found that only a small percentage of those reporting a positive intention to vaccinate followed through on receiving the vaccine after two months [54]. 

### Limitations and Further Considerations

Our findings and their interpretations are subject to a number of limitations and considerations. We acknowledge that there is a bias in this body of literature towards settings where vaccines have been rolled out, studied and reported. Settings that lack reliable data collection about COVID-19 vaccinations are less likely to be represented in published academic literature, and therefore less likely to be included in this scoping review. 

Across studies, approaches to defining COVID-19 vaccination coverage were not standardised. For the purpose of our scoping review, we adopted a broad definition for the COVID-19 vaccination coverage indicator and included studies reporting on receipt (or non-receipt) of a single dose, multiple doses and/or booster doses. Nearly one in five of the articles included in our review reported on more than one vaccination coverage indicator that met our inclusion criteria. The application of common definitions for COVID-19 vaccination coverage would facilitate greater cross-study comparability and more nuanced analyses. 

We relied on the PROGRESS-Plus framework as a starting point to guide how we grouped and labelled dimensions of inequality. Alternate frameworks may have yielded different conclusions about the most frequently reported dimensions of inequality. We did not report political factors as relevant dimensions of inequality, although we noted that 13 of the 167 articles reported on inequalities based on political views or voting patterns. Of these studies, 11 were conducted in the United States, all of which found lower vaccination among Republican voters and/or higher vaccination among Democrat voters. 

Our exploration of inequality trends for selected dimensions of inequality in this scoping review was premised on findings that may be of variable quality. Approaches and thresholds to determine the ‘meaningfulness’ of inequality were different across studies. More rigorous meta-analyses incorporating quality assessments are required as an extension of our initial findings. 

As per the design of our scoping review, we did not account for how countries may have prioritised different populations during phased vaccine rollouts. Initially, COVID-19 vaccine doses were in limited supply and inequality during the early stages of their distribution was inevitable (though COVID-19 vaccination coverage equity remains an end goal for most countries) [10]. Many of the included articles, however, did take this into account in their study design. We did not focus on the reasons underlying vaccination status, such as whether population subgroups remained unvaccinated by choice (low acceptance of the vaccine) or their circumstance (low access to the vaccine). Explorations of the drivers of inequality were outside the scope of this review. We did not differentiate between studies conducted in general populations versus studies that evaluated a specific campaign or programme, which may have been targeted towards certain populations. 

## 5. Conclusions

In this scoping review, we assessed 167 research articles to provide an overview of how within-country COVID-19 vaccination coverage inequality has been studied. Our findings demonstrate that most research to date has been conducted in higher income countries, underscoring the need for expanded research in other contexts to gain a fuller understanding of patterns of inequalities across populations and settings. While we characterised research on diverse dimensions of inequality, those most frequently studied were related to demographic factors. The trends that we reported for inequalities by age, race/ethnicity/cultural group/language/nationality/country of birth, and sex/gender dimensions of inequality were intended to be preliminary and exploratory. More detailed analyses across these and other dimensions of inequality are warranted, including dedicated systematic reviews and meta-analyses to draw more reliable and specific conclusions. Research in this topic area can be further strengthened by adopting standardised COVID-19 vaccination indicators, which would promote greater cross-study comparability.

As COVID-19 vaccination programmes, including the administration of booster doses, continue to expand globally, ongoing efforts are needed to grow this body of research and capture the evolution of inequalities in vaccination coverage, both globally and locally within countries. The characterisation of inequalities related to multiple, diverse dimensions of inequality (encompassing both context-specific and universally applicable dimensions) stands to offer relevant lessons and insights for strengthening equity in vaccine policies, planning and implementation.

## Figures and Tables

**Figure 1 vaccines-11-00517-f001:**
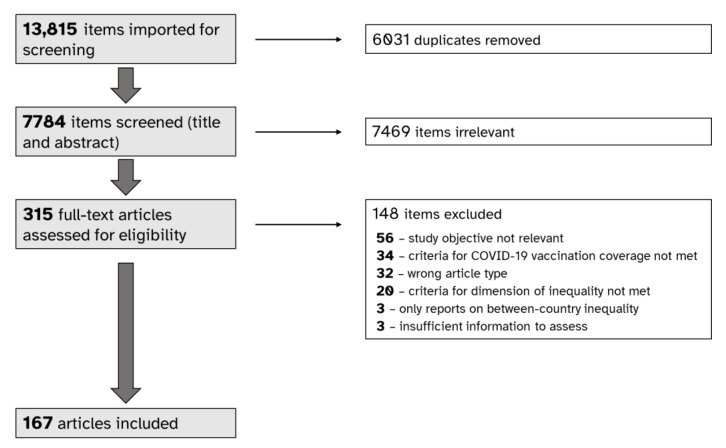
PRISMA flow diagram depicting stages of selecting sources for a scoping review about inequalities in COVID-19 vaccination coverage.

**Table 1 vaccines-11-00517-t001:** Criteria to determine relevance of COVID-19 vaccination coverage indicator for a scoping review about inequalities in COVID-19 vaccination coverage.

Inclusion Criteria	Exclusion Criteria
Indicator captures receipt or non-receipt of COVID-19 vaccine and/or booster dose May specify any number of vaccine or booster doses May be self-reported, reported by a proxy (such as a parent) or obtained through administrative source or health records	Indicator captures: ○ intention to vaccinate ○ attitudes about vaccination ○ vaccine availability, accessibility or eligibility ○ access to vaccination sites ○ vaccine readiness ○ vaccine decision-making factors ○ participation in vaccination trials ○ strategies for increasing vaccination uptake ○ predictive modelling of vaccination uptake ○ perceptions about vaccination uptake

**Table 2 vaccines-11-00517-t002:** Countries represented by one or more study included in a scoping review about inequalities in COVID-19 vaccination coverage.

Country Income Group Classification	Countries
High income	Australia (2 articles); Canada (6 articles); Czech Republic (1 article); France (2 articles); Germany (1 article); Greece (2 articles); Hong Kong (5 articles); Hungary (2 articles); Israel (7 articles); Italy (2 articles); Japan ^a^ (2 articles); Kuwait ^a^ (1 article); Latvia (1 article); Netherlands (1 article); New Zealand (1 article); Norway ^a^ (1 article); Qatar ^a^ (1 article); Romania (1 article); Saudi Arabia ^a^ (3 articles); Singapore (2 articles); Slovakia ^a^ (1 article); Spain (1 article); Sweden ^a^ (4 articles); United Arab Emirates ^a^ (2 articles); United Kingdom ^a^ (15 articles); United States of America ^a^ (86 articles)
Upper-middle income	Belarus (1 article); Brazil (2 articles); China (3 articles); Guatemala ^a^ (1 article); Iraq ^a^ (1 article); Jordan ^a^ (1 article); Kazakhstan ^a^ (1 article); Mexico ^a^ (3 articles); Peru (1 article); Serbia (1 article); Thailand (1 article)
Lower-middle income	Bangladesh (1 article); Egypt (1 article); India ^a^ (3 articles); Indonesia (1 article); Lebanon ^a^ (1 article); Pakistan (1 article)
Low income	Ethiopia (1 article); Guinea (1 article); Malawi (2 articles)
Not classified	Palestine ^a^ (3 articles)

^a^ Country included in at least one multi-country study or review article.

**Table 3 vaccines-11-00517-t003:** Dimensions of inequality featured in sources for a scoping review about inequalities in COVID-19 vaccination coverage, including corresponding number of articles, percentage of total number of articles (*n* = 167) and examples of measurement criteria.

Dimension of Inequality	Number of Articles (% of Total) ^a^	Illustrative Examples of Measurement Criteria ^b^
Age	127 (76%)	years; parental age; above or below median age of population
Race, ethnicity, cultural group, language, nationality or country of birth	117 (70%)	White, Black, Hispanic, Asian or Other (applicable to studies based in the United States); language at home; national or foreigner
Sex or gender	103 (62%)	male or female; transgender or non-binary (yes/no)
Occupation- or employment-related factor	81 (49%)	employed or unemployed; employment in healthcare industry (yes/no); essential worker, non-essential worker, or non-working status; public or private sector employee; military ranking; profession
Education level	76 (46%)	years of schooling; highest level of schooling completed; highest qualification
Subnational region or area	68 (41%)	state/province; region; municipality; census tract; county; health zone
Economic status	68 (41%)	household income; above or below defined poverty line; self-perceived financial status; level of difficulty covering household expenses
Place of residence	39 (23%)	urban or rural; metro or non-metro; urban, rural or camp; population size of zip code
Vulnerability, deprivation or poverty index	38 (23%)	Social Vulnerability Index (applicable to studies based in the United States); Index of Multiple Deprivation (applicable to studies based in the United Kingdom); Human Development Index; Multi-Dimensional Poverty Index
Marital status	30 (18%)	single, married, cohabitating, divorced, widowed; living with partner (yes/no)
Family size or composition	27 (16%)	number of children; household size; elderly living with family (yes/no); children living in household (yes/no); living alone (yes/no)
Health insurance	27 (16%)	insured or uninsured status; private or public health insurance type
Disability status	10 (6%)	self-reported living with a disability (yes/no); extent of daily activity limitation
Religion	8 (5%)	religious affiliation (e.g., Christian, Buddhist, Hindu, Jewish, Muslim, Sikh, other, none)
Housing type or characteristic	7 (4%)	homeless (yes/no); house owned, private rented, social rented, other; type of housing: mobile, detached house, attached house, multiunit apartment, etc.
Migration status	5 (3%)	citizen, landed immigrant, refugee, temporary/other; migration history (yes/no)
Social capital	3 (2%)	trust in others; civic participation; social capital index
Sexual orientation	3 (2%)	bisexual, gay/lesbian, heterosexual
Other: food security, incarceration status, income inequality, car ownership, computer ownership, school type	1–2 each	

^a^ The scoping review included a total of 167 articles; most articles featured more than one dimension of inequality. ^b^ Note that this is not an exhaustive list of all approaches to measuring the dimensions of inequality.

## Data Availability

A list of the articles included in the review is available upon request from the authors.

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
