# Peer review of "Within-Country Inequality in COVID-19 Vaccination Coverage: A Scoping Review of Academic Literature"

_vaccines, 2023, doi:10.3390/vaccines11030517_

Round 1

Reviewer 1 Report

This paper describes a scoping review of research articles investigating possible within-country inequalities in COVID-19 vaccination coverage. Characteristics of these articles, their methods and findings are described. The paper does not aim to meta-analyse the results of the articles described.

Major comments: none.

Minor comments:

1) l. 178-180 are probably part of a template, and should be deleted?

2) l. 240 – 242: The formulation of the numbers here is not clear, while 84+84+1 = 169 and not 167.

3) l. 293-295:  “…..the majority (89 articles, or 70%) found higher vaccination among older groups and/or lower vaccination among younger groups.”
l. 347:  “…..vaccination was higher among males or lower among females….”
It is unavoidable that if vaccination is higher in old ages, it is lower in young ages, similar for males/females. Removing superfluous text will improve readability.

4) l. 297 -300: “For example, a study by Tessier et al (2022) assessing non-receipt of the COVID-19 vaccine among older adults in England reported the odds of being unvaccinated were higher for the younger end of the age range, both among those aged 50-69 years, and among those aged 70 or more [22].”
The sentence is rather complicated and the statement remains unclear.

Author Response

1) l. 178-180 are probably part of a template, and should be deleted?

Thank you. They have been deleted.

2) l. 240 – 242: The formulation of the numbers here is not clear, while 84+84+1 = 169 and not 167.

We agree that this could be written more clearly. There were 83 studies that used survey data and 83 studies that used administrative or surveillance records, plus 1 study that used both (83 + 83 + 1 = 167).

The one study that used both sources is included in the counts for each of the sources (84 + 84).

We have revised this sentence as: “Information about COVID-19 vaccination coverage was sourced from surveys (84 articles), and administrative or surveillance records, including health records (84 articles), noting that one article used data from both of these types of sources.”

3) l. 293-295:  “…..the majority (89 articles, or 70%) found higher vaccination among older groups and/or lower vaccination among younger groups.”
l. 347:  “…..vaccination was higher among males or lower among females….”
It is unavoidable that if vaccination is higher in old ages, it is lower in young ages, similar for males/females. Removing superfluous text will improve readability.

We have removed the superfluous text as suggested to improve readability.

4) l. 297 -300: “For example, a study by Tessier et al (2022) assessing non-receipt of the COVID-19 vaccine among older adults in England reported the odds of being unvaccinated were higher for the younger end of the age range, both among those aged 50-69 years, and among those aged 70 or more [22].”
The sentence is rather complicated and the statement remains unclear.

We have clarified the sentence as follows: “For example, a study in England reported higher rates of being unvaccinated in younger individuals of study populations aged 50 years or older [22].

Reviewer 2 Report

very interesting and well written scoping review. Limitations are correctly reported by the authors and I think that there are no areas of weakness

Author Response

Thank you for taking the time to review our submission.

Reviewer 3 Report

The article presented for review is very interesting and concerns an important issue, which is the conditions of vaccination against COVID-19.

The work has a very important methodological character of a scoping review of academic literature. Analyzes performed in this way, based on the correct search criteria, provide a large amount of information.

The procedure used in the work has been described in detail and clearly (the suplementary materials are very useful) and the software used is correct and adequate.

The work contains interesting research results. The authors initially identified and described 3 main dimensions: 3.3.1. age; 3.3.2. Race, ethnicity, cultural group, language, nationality or country of birth and 3.3.3. sex or gender. These are very interesting results, and certainly require a broader discussion.

The remaining "Dimensions of inequality" seem to require further analysis - education and occupation are important factors in many works. Therefore, I consider the lack of such an in-depth analysis to be the only (small) shortcoming of the reviewed article.

Author Response

Thank you for taking the time to review our submission.

Regarding your comment about in-depth analyses of additional dimensions of inequality: We agree that further analysis of other dimensions of inequality such as education and occupation – as well as more rigorous analysis of age, race/ethnicity and sex/gender – are important extensions of this work. These further analyses will benefit from meta-analysis methods (which were not within the scope of our paper).